# Home Birth in Portugal—A Comprehensive Analysis Based on Official Statistical Data

**Sónia Pintassilgo [1,\*], Mário J. D. S. Santos [1,2] [ID], Inês Trindade [1] and Dulce Morgado Neves [1]**

[1] ISCTE—Instituto Universitário de Lisboa, Centro de Investigação e Estudos de Sociologia (CIES-Iscte), Avenida das Forças Armadas, 1649-026 Lisboa, Portugal

[2] Department of Sociology, Universidade da Beira Interior, Estrada do Sineiro, 6200-209 Covilhã, Portugal

\* Correspondence: sonia.cardoso@iscte-iul.pt

**Abstract:** To date, there are no comprehensive analyses of the official data on home births in Portugal that consider other determinants beyond place of birth, such as the conditions in which births occur or the mothers' social profiles. The aims of this article were: (1) to describe the characteristics of childbirth in Portugal according to the place of birth, and (2) to analyse the association between home birth assistance and perinatal, neonatal, and infant mortality rates. We performed a descriptive, correlational analysis of the official datasets of live births in Portugal (census-based) produced by the National Institute of Statistics. In 2020, home births had more similarities with hospital births than to births occurring in any "other place". Furthermore, considering the 1995–2020 timeframe, home births with specialised assistance were negatively correlated with perinatal, neonatal, and infant mortality, and home births with non-specialised professional assistance or those that were unassisted showed a positive correlation with these mortality rates. Home births are heterogenous, and merely referencing the place of birth does not provide enough information. Although it recommended further investigation, this analysis pointed to the need to assure a specific, systematic evaluation of the quality of care in home births that allows for the consistent assessment and improvement of its safety.

**Keywords:** home childbirth; perinatal mortality; infant mortality; neonatal mortality; maternity care; social analysis; demographic analysis

## 1. Introduction

Compared to other European countries, Portugal had a late childbirth hospitalisation movement. In 1960, approximately 80% of all births still occurred at home, but by 1985, home births were already rare (Santos 2018). In the early 20th century, when the first maternity wards were created in the three main Portuguese cities, a hospital was not consensually considered the most adequate place to give birth, even among medical doctors. As in other European countries, the perinatal health outcomes associated with hospital births were generally poorer than those associated with home births; thus, many doctors advocated for improving the quality of childbirth care at home instead of promoting hospitals (Baptista 2016). By then, while some home births had the professional assistance of a midwife or a doctor, most were informally attended by lay midwives or by older and experienced female relatives or neighbours (Carneiro 2008). In fact, the high rates of home births were also sustained by a prevalent moral belief—subscribed and promoted by the state—that women should stay at home with their families (Baptista 2016; Carneiro 2008). Until approximately the mid-20th century, most pregnant women had little or no antenatal care, and hospital wards for maternity care were mostly dedicated to the care of unprivileged women from urban areas (Carneiro 2008).

Particularly from the 1950s onward, the criticism over the poor organisation of maternity care in the country grew, fuelled by the persistent high rates of infant and maternal

mortality. This led to reframing the organisation of maternity care, slowly making antenatal and intrapartum care available for all women, placing childbirth—even if unproblematic—under the control of medicine, and inscribing the hospital as the safest place to give birth. The rapid decrease in infant mortality rates following the mainstreaming of hospital childbirths legitimised and reinforced the general acceptance of the hospital management of childbirth, and it definitely outcasted home births and home birth carers. The hospital represented the modern solution—the future—while home birth represented the memory of an underdeveloped past (Fedele and Guignard 2018; Santos 2017; Vallgårda 2012).

Currently, having or attending a home birth in Portugal is legal, even if there is no legislation that specifically addresses this option or practice of birth. (Law 15/2014) generically defines that "each health services user has the right to choose the services and the care providers, within the available resources", but so far, this has had little or no practical translation to home birth care. Home birth is an available option for those with private funding as it is paid out-of-pocket by families. Some families with private health insurance that covers "nursing care at home" have successfully claimed a partial reimbursement of their expenses, but there is no insurance that clearly covers home birth care.

In this sense, contemporary planned home births in Portugal—births that intentionally take place at home—are distinct from mid-20th century home births in terms of the sociodemographic characteristics of the parents (Pintassilgo and Carvalho 2017), their motivations, and their use of technologies (Santos and Augusto 2016), as well as in terms of the knowledge and training of the professionals involved. Yet, this option is still condemned by many today under the belief that it will bring back the rural, poor, underdeveloped, and unwanted past, strongly increasing maternal and perinatal mortality and undoing the progresses made in terms of maternal and child healthcare (Rocha 2016; Santos 2014).

To date, however, little is known about the characteristics of contemporary home births in Portugal. As in other countries in Europe, Portuguese statistical data from official sources do not distinguish between planned or unplanned home births—not to mention cases of mis- and under-reporting—making any in-depth, population-based study of these issues challenging (Galková et al. 2022). Hence, home births remain generally unknown and invisible to the general population, to regulatory bodies, and to the government, limiting the scope of any public debate on these matters, which is frequently anchored in prejudice and personal opinions rather than based on science (Santos 2014).

It is noteworthy that even in countries where data on birth outcomes by place of birth are available, the existing literature does not always detail the conditions in which these births take place, and it is as if they were homogeneous or essentially defined by the place itself—home or hospital. The analyses tend to focus, above all, on trends in the evolution of home birth (MacDorman et al. 2010), on the analysis of specific regions or populations (Gregory et al. 2021), on ethical issues and decision-making about the place of birth (Minkoff and Ecker 2013; de Vries et al. 2013), or on risk management and health outcomes (Grünebaum et al. 2020).

When considering the same place of birth, finding important differences in maternal and perinatal health outcomes between countries or regions points to the importance of considering other determinants beyond place of birth, such as the conditions in which births occur, namely, in terms of the planning and type of assistance (Grünebaum et al. 2020; Reitsma et al. 2020; Santos 2018), as well as in terms of the parents' social profiles. In this sense, previous studies have reported that different profiles associated with childbearing, such as age, education, and marital status, are associated with different care profiles (Hildingsson et al. 2010; Pintassilgo and Carvalho 2017). There is some evidence that planned home births tend to be carried out by people who are better educated, healthier, and more satisfied with care compared to those who have planned hospital caesarean sections (Hildingsson et al. 2010). The correspondence between the social profiles of parents, the types of assistance, and the health outcomes in each place of birth deserves to be further explored for better understanding both the international variations and country-

level results, particularly in the context of the limited integration of data. Thus, analysing home births by the types of assistance and the characteristics of the parents may allow for a tentative distinction between planned and unplanned births at home, and it may offer a more detailed portrait of this place of birth.

As such, the aims of this article are, on the one hand, to describe the characteristics of childbirth in Portugal according to the place of birth, and on the other hand, to analyse the association between the type of home birth assistance and perinatal, neonatal, and infant mortality rates. The data referring to maternal mortality rate were left out of this analysis. It should be noted that data on maternal deaths are always difficult to analyse using a quantitative approach since the analysis deals with a reduced amount of data. Considering maternal mortality as a potentially avoidable phenomenon (Loudon 1992; Pintassilgo 2014), each maternal death would deserve a qualitative analysis at its base, which was outside the scope of this study. This analysis focused on the Portuguese case, a country where there was an increase in home birth rates in 2020 compared to previous years. The pandemic context may have contributed to this increase, considering that guidelines and procedures, both for hospital births (Davis-Floyd et al. 2020) and home births (Romanis and Nelson 2020; Dahlen et al. 2020), underwent important changes in different countries and regions.

## 2. Materials and Methods

The sources of information used were the official Portuguese databases and statistics on birth and maternal and child health (anonymised secondary data), produced by the National Institute of Statistics (INE) of Portugal, in accordance with the principles of the National Statistical System, regulated by Law no. 22/2008.

The live births database produced by INE has resulted from the mandatory registration of live births, up to 20 days after each birth, in civil registry offices. At the time of registration, an official INE questionnaire (verbete de nado-vivo) is filled out with information on various characteristics of the birth and the newborn, parents, and birth attendance. After completion, the questionnaires are sent from the civil registry offices to the INE, allowing the annual construction of official databases on birth and the calculation of different indicators, and this information is published in the Demographic Statistics (official statistics). Users can access the indicators produced and published in Demographic Statistics, or they can request access to the databases, in order to autonomously carry out their own calculations and analyses, according to the objectives of their respective research.

Thus, a census-based database of live births for the year 2020 was used within the scope of the existing protocol between INE and the Ministry of Science and Higher Education, which allows accredited researchers to access official statistical information made available through microdata. A descriptive statistical and demographic analysis was performed using the statistical software SPSS v.27. From the calculation of proportions in relation to the total number of live births or to the total of live births in each place of birth, the analysis considered the frequency of live births according to the place of birth—home, hospital, other place (births that took place neither at home nor at the hospital)—and the distinction of the characteristics of these events according to those categories. The characteristics identified were relative to different dimensions of analysis: characteristics of mothers ('mother's education' and 'mother's nationality and residence'), characteristics of pregnancy (length of pregnancy' and 'parity'), and the type of care ('birth care provider'). The descriptive analysis about birth conditions considered the individual cases of births officially registered in Portugal in 2020. This analysis will allow for describing and identifying possible differences in the characteristics of the people who gave birth in Portugal by place of birth.

The second part of this paper considers the results of health indicators produced by the INE and the Directorate General for Health (DGS) that were published in demographic statistics or health statistics for the period 1995 to 2020, namely, perinatal, neonatal, and infant mortality.

The perinatal mortality rate is the ratio of fetal deaths (28 weeks or more of gestation) occurring in the first week of life for live births. The neonatal mortality rate is calculated by dividing deaths during the first month of life by the total number of live births. Infant mortality rates result from the quotients between infant deaths (up to the first year of life) and live births. All rates were expressed as per thousands. The rates shown were those published for one-year periods in the official statistics.

Furthermore, we analysed the correlation between the frequency of home births and the perinatal, neonatal, and infant mortality rates over a period of more than two decades. To investigate the correlation between mortality rates and home births, groups were created according to the type of home birth: home birth with a specialised care provider (doctor, midwife, or obstetric nurse), home birth with a non-specialised care provider (nurse or other), and home birth without any care provider.

The analysis was based on the verification of the correlation coefficient between the indicators and the three different types of home births, which consisted of a correlation measurement between the variables through a Pearson's coefficient (r). Thus, we analysed the association between home birth care provider and perinatal, neonatal and infant mortality rates.

## 3. Results

### 3.1. Place of Birth in Portugal between 1995 and 2020

The proportion of live births by place of birth in Portugal revealed very low rates for home births for several decades. From 1995 to 2020, the rate of home births was always lower than 1% (Figure 1), except for three years—at the beginning of this period, in 1995 and 1996, and, more recently, in 2020.

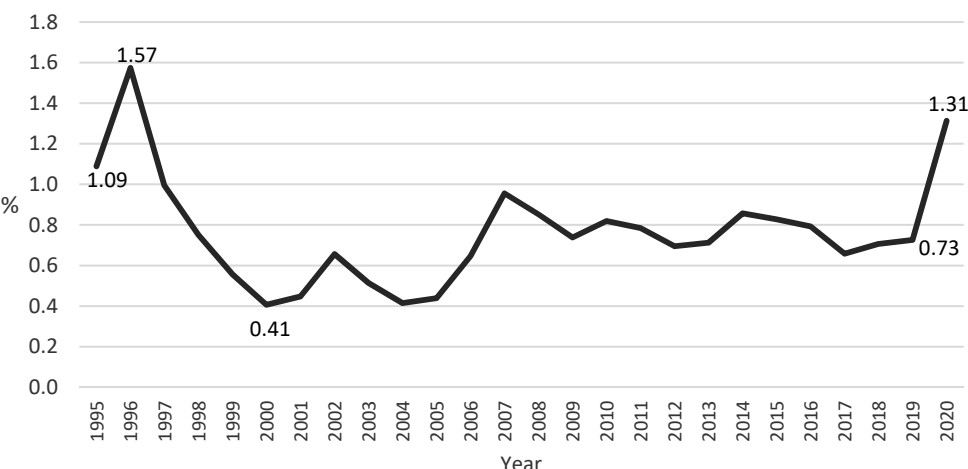

**Figure 1.** Proportion of home live births per total of live births (%) in Portugal for the period 1995–2020 (based on data from the live births database, INE 2020).

In 2020, the proportion of home births grew compared to previous years, and it represented 1.3% of the total of live births. The pandemic context and the restrictions imposed by health authorities may have been some of the explaining factors for this increase (Barata et al. 2020). Still, home births were less than the number of live births occurring in other places (1.6%), and the majority of births (97.1%) took place at the hospital (Table 1).

Despite the small proportion of live births at home, it was important to further analyse these events (Santos 2020; Galková et al. 2022) and compare their characteristics with those of live births occurring in hospitals and other places.

**Table 1.** Live births (N. and %) by place of birth in Portugal, 2020 (based on data from the live births database, INE 2020).

| Place of Birth | N. | % |
|---|---|---|
| Home | 1112 | 1.3 |
| Hospital | 82,205 | 97.1 |
| Other place | 1374 | 1.6 |
| Total | 84,691 | 100 |

### 3.2. Home and Hospital Births in Portugal in 2020

For the year 2020, we focused on the distinction between live births at a hospital, at home, and at other places in terms of the characteristics of the mothers, pregnancy, and birth assistance, and the results are shown in Table 2.

**Table 2.** Live births (%) by mothers, pregnancy, and care characteristics un Portugal in 2020 (based on data from the live births database, INE 2020; ISCED-2011, International Standard Classification of Education).

| Dimension | Variable | Category | Place of Birth | | |
|---|---|---|---|---|---|
| | | | **Home** | **Hospital** | **Other Place** |
| Mothers | Mother's educational level | Until primary | 7 | 6.8 | 2.1 |
| | | Lower secondary | 16.3 | 15.1 | 4.7 |
| | | Upper secondary | 31.9 | 33.7 | 8.4 |
| | | Tertiary | 36.2 | 37.3 | 6.8 |
| | | Unknown | 8.6 | 7 | 77.9 |
| | Nationality | Portuguese | 84.1 | 86.4 | 94 |
| | | Other | 15.9 | 13.6 | 6 |
| Pregnancy | Length of pregnancy (in weeks) | Until 35 | 3.5 | 3.9 | 1.3 |
| | | 36 | 3.2 | 2.8 | 0.5 |
| | | 37 | 6.7 | 7.3 | 1.5 |
| | | 38 | 17.8 | 18.2 | 3.6 |
| | | 39 | 34 | 33.3 | 8.6 |
| | | 40 | 25.8 | 25.1 | 6 |
| | | 41 | 7.9 | 8.7 | 1.5 |
| | | 42 or more | 0.4 | 0.2 | 0.1 |
| | | Unknown | 0.8 | 0.3 | 76.9 |
| | Parity | 1 | 50.7 | 53.6 | 87.6 |
| | | 2 | 33.5 | 33.4 | 9 |
| | | 3 | 11.1 | 9.6 | 2.8 |
| | | 4 or more | 4.7 | 3.3 | 0.7 |
| Assistance | Care provider | Doctor | 63.1 | 75.4 | 15.9 |
| | | Midwife | 28.9 | 24.2 | 4.9 |
| | | Nurse | 1.3 | 0.3 | 0.5 |
| | | Other | 3.4 | 0 | 1.2 |
| | | No care provider | 3 | 0 | 0.4 |
| | | Ignored | 0.4 | 0 | 77.1 |

### 3.2.1. Sociodemographic Characteristics of the Mothers

Approximately 68% of the mothers who gave birth at home had high levels of education, with school qualifications corresponding to tertiary education (36%) and upper-secondary education (32%). Women whose children were born in hospitals had similar profiles, with slightly higher values in the two mentioned categories (71% in total). Women

with primary education, lower secondary education, or unknown education were less represented for both the home and hospital births. On the other hand, for most of the women who gave birth in other places (78%), there was no information about their scholarly qualifications, and women with upper-secondary education composed the second most represented category (less than 10%).

### 3.2.2. Nationality and Residence of the Mother

It is for home births that the highest proportion of non-native women was recorded (approximately 16%), which was slightly above the proportion registered for hospital births. Live births in other places registered the lowest proportion of women from a foreign country.

The group of women who gave birth at home included 40 different nationalities from four continents, with half of the countries belonging to Europe (Table 3). The nationalities of the women from foreign countries who had live births at home were mostly European and American, with each continent representing 37.9% of the cases, followed by Africa (with 15.8% of cases) and Asia (with 8.4%). In this sense, births for mothers from Europe made up a much higher percentage of the home births (37.9%) than the hospital births (16.5%). On the other hand, births for mothers from Asia made up a much lower proportion of the home births (8.5%) than the hospital births (15.2%). Considering the distribution by country, the most-represented nationality for home births was Brazilian (31.1%), followed by German (10.2%) and Cape Verdean and Italian (both with 4.0%). As for the hospital births, these concerned mothers from all continents whose nationalities were primarily Brazilian, Angolan, Ukrainian, and Nepalese.

**Table 3.** Countries and live births (N. and %) by place of birth and foreigners' mothers' nationalities in Portugal in 2020 (based on data from the live births database, INE 2020).

| Mother's Continent (Nationality) | Place of Birth | | | | | | | | | |
|---|---|---|---|---|---|---|---|---|---|---|
| | Home | | | | | Hospital | | | | |
| | Countries (N.) | Countries (%) | Live Births (N.) | Live Births (%) | Most Represented Country | Countries (N.) | Countries (%) | Live Births (N.) | Live Births (%) | Most Represented Country |
| Africa | 9 | 22.5 | 28 | 15.8 | Cape Verde | 40 | 28.4 | 3065 | 27.4 | Angola |
| America | 5 | 12.5 | 67 | 37.9 | Brazil | 44 | 31.2 | 4558 | 40.8 | Brazil |
| Asia | 6 | 15.0 | 15 | 8.5 | Nepal | 28 | 19.9 | 1701 | 15.2 | Nepal |
| Europe | 20 | 50.0 | 67 | 37.9 | Germany | 27 | 19.1 | 1848 | 16.5 | Ukraine |
| Oceania | 0 | 0.0 | 0 | 0.0 | | 2 | 1.4 | 3 | 0.03 | N. Zealand |
| Total | 40 | 100 | 177 | 100 | | 141 | 100 | 11,175 | 100 | |

Regarding the mothers' residences, in the case of the live births at home, the highest proportion was located in the municipality of Lisbon, the capital, and Sintra, a municipality also belonging to the metropolitan area of Lisbon.

### 3.2.3. Length of Pregnancy

The length of pregnancy had similar distributions for both home and hospital births (Table 2), with most births occurring at 39 weeks, followed by births occurring at 40 weeks. Even so, home births tended to happen slightly later (13.4% occurred up to 37 weeks, and 85.9% occurred from 38 weeks onward) than hospital births (14% occurred up to 37 weeks, and 85.5% occurred from 38 weeks onwards).

### 3.2.4. Parity

Regardless of the place of birth, first children had the highest proportion because most of the mothers had no previous children. It was, however, evident that of the total number of live births that occurred in each place, the proportion of live births in order one (that is, those corresponding to the first children) was higher in the hospital setting (compared to the proportion of first children born at home), and the proportion of order two and later births was higher at home (Table 2).

Concerning births occurring in other places, the vast majority (approximately 88%) corresponded to first children. This concentration did not result from a greater proportion of live births for mothers of younger ages (compared to those occurring at home or in a hospital). However, it was associated with a majority of births occurring outside of marriage, with cohabitation (45%) or without cohabitation (19%), and these were higher figures than those found for home and hospital births.

### 3.2.5. Birth Assistance

In an overall reading, one could say that in 99.9% of hospital births and in 93.3% of home births, care was provided by a health professional (doctor, midwife, or nurse).

Thus, the hospital is the place of birth with the least dispersion in terms of the type of assistance since all cases were concentrated in three categories (doctor, midwife, or nurse) and a majority of the births (more than two thirds) were assisted by doctors (Table 2). Contrary to what was registered for "other place", in which the "ignored care provider" prevailed, home births were also, for the most part, registered as having been assisted by doctors.

In fact, for home births, official data revealed that assistance had been more frequently provided by doctors, to the detriment of midwives who, in the last decade, and particularly, in recent years, have appeared to assume a smaller proportion of participation in home birth assistance in Portugal (Figure 2). In 2020, less than 30% of live births were attended by midwives.

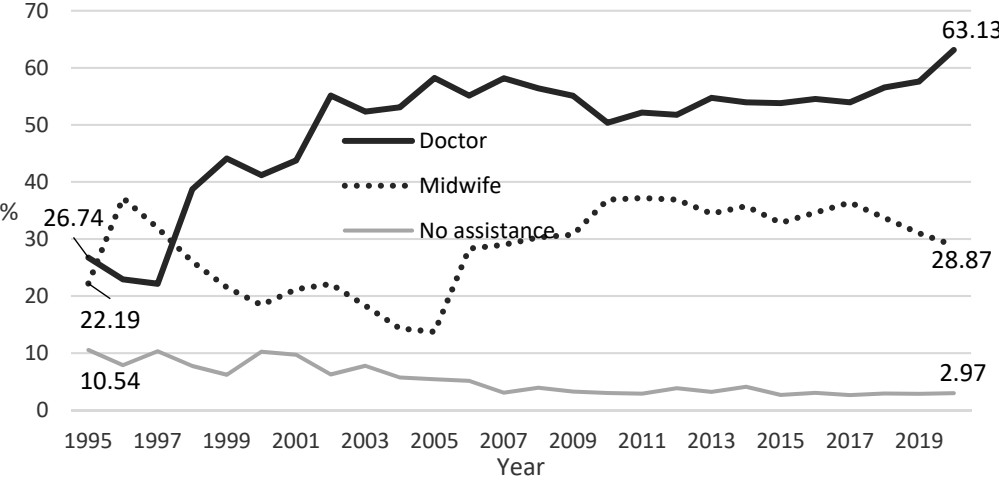

**Figure 2.** Proportion of home live births by care provider (%) in Portugal for the period 1955–2020 (based on data from INE, Demographic Statistics, 1995–2020).

### 3.3. Home and Hospital Births in Portugal in 2020

Figure 3 follows the evolution of the frequency of live births at home and of the perinatal mortality rates in Portugal. In the last two-and-a-half decades, the relationship between both phenomena was not evident, and it deserves a deeper analysis.

The analysis of home births between 1995 and 2020 and infant, perinatal, and neonatal mortality rates for those same years revealed that there was no significant correlation between the place of birth and infant, perinatal, and neonatal mortality rates. However, subdividing home births into three distinct groups—home births with skilled care providers, home births with non-skilled care providers, and unassisted home births—we found some significant correlations, namely, the infant, perinatal, and neonatal mortality rates that were presented. Regarding home births that occurred with specialised care providers, there were significant negative correlations between the variables, as shown in Figure 4. Infant, perinatal, and neonatal mortality rates decreased when there were higher numbers of home births with this type of assistance. There was a high negative linear correlation (Cohen 1988) between births in the home with specialised care providers and the infant mortality

rate (r = −0.921, *p* = 0.000), and the same was true for perinatal (r = −0.893, *p* = 0.000) and neonatal mortality rates (r = −0.897, *p* = 0.000).

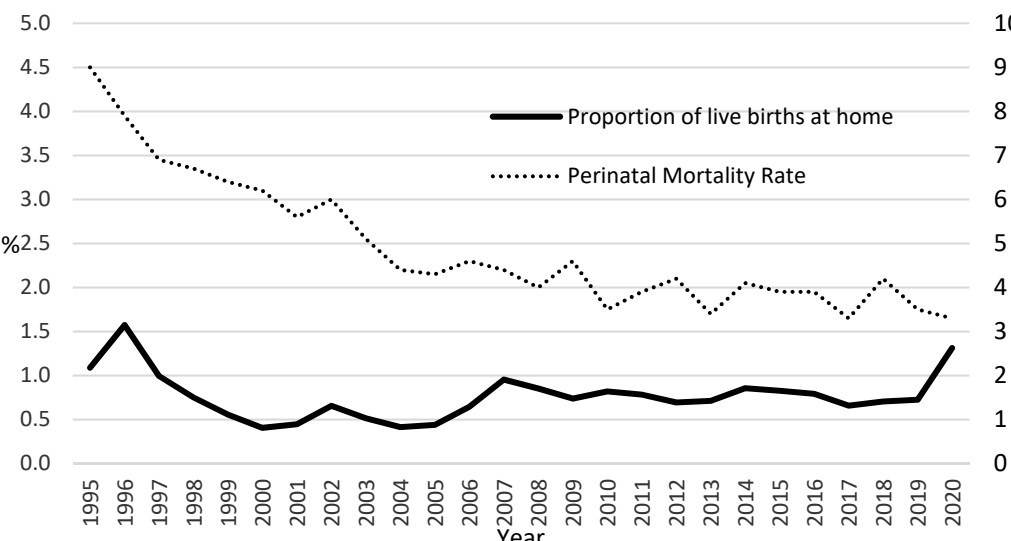

**Figure 3.** Proportion of live births at home (%) and perinatal mortality rates (per thousand) in Portugal for the period 1995–2020 (based on data from INE, Demographic Statistics, 1995–2020).

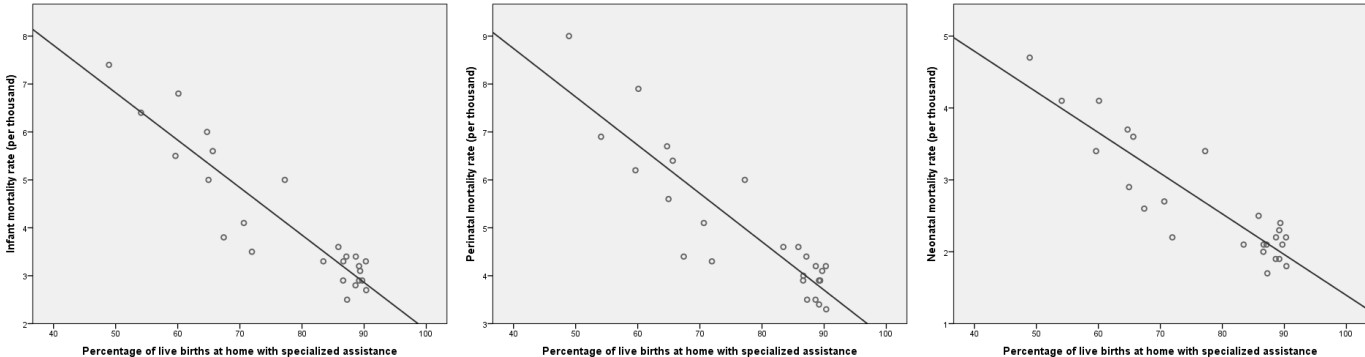

**Figure 4.** Representative graphs of the correlations between home live births with skilled care providers and infant mortality rate, perinatal mortality rate, and neonatal mortality rate.

However, when analysing the correlations between home births with non-specialised care providers and infant, perinatal, and neonatal mortality rates (Figure 5), we saw a high positive linear correlation (Cohen 1988) between births at home with non-specialised care providers and infant mortality rate, with an increase in the value of one variable when the other increased (r = 0.887, *p* = 0.000). The same was true for perinatal (r = 0.895, *p* = 0.000) and neonatal mortality rates (r = 0.873, *p* = 0.000).

There were also very similar results from the analysis of the correlations between unattended home births, with significant positive linear correlations between this type of birth and infant, perinatal, and neonatal mortality rates (Figure 6). When analysing the Pearson's correlation, there were also high positive linear correlations (Cohen 1988) between unattended home births and infant (r = 0.882, *p* = 0.000), perinatal (r = 0.855, *p* = 0.000), and neonatal mortality rates (r = 0.850, *p* = 0.000).

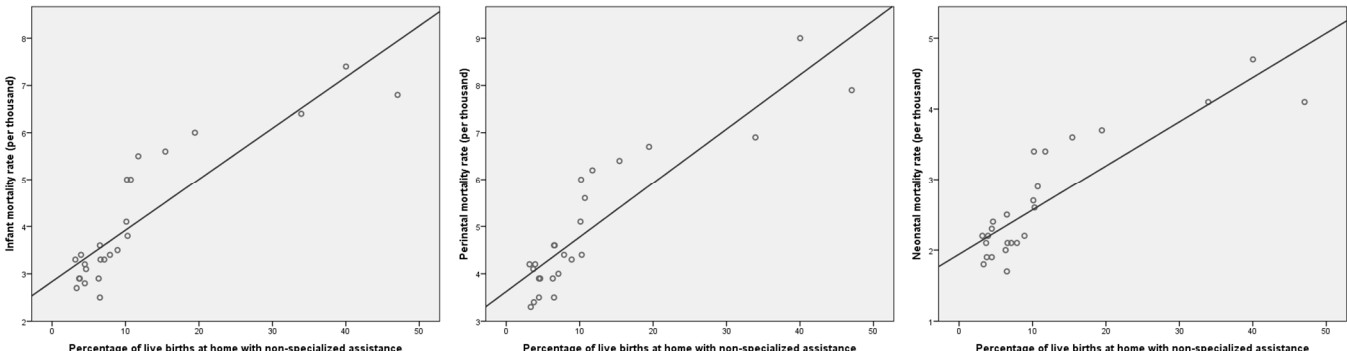

**Figure 5.** Representative graphs of the correlations between home live births with non-specialised care providers and infant mortality rate, perinatal mortality rate, neonatal mortality rate.

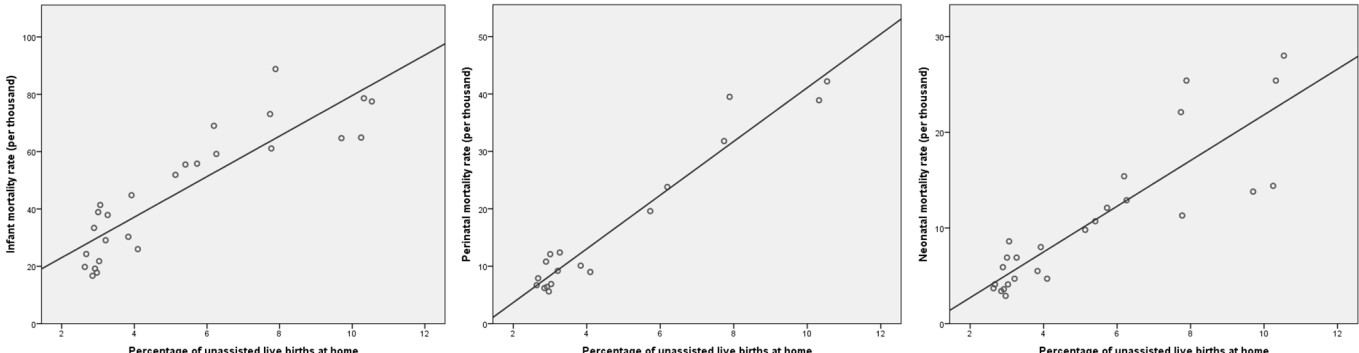

**Figure 6.** Representative graphs of the correlations between unassisted live home births and infant mortality rate, perinatal mortality rate, and neonatal mortality rate.

## 4. Discussion

As far as the analysis revealed considering the place of birth was not enough to describe and understand home births. The same could be said regarding hospital births. As earlier demonstrated by Loudon (1992), it was not so much the place but the quality of the staff and the circumstances in which care was provided that had a greater influence on the overall quality and safety of home or hospital childbirth care.

### 4.1. Place of Birth: Characteristics of Mothers, Pregnancy, and Type of Care

Previous studies have pointed to the idea that different profiles of mothers (in particular, with regard to their qualifications) lead to different care profiles (Hildingsson et al. 2010; Pintassilgo and Carvalho 2017). As such, in the present study, the mothers' sociodemographic profiles revealed similarities for both contexts (hospital births and home births) in terms of education level. The highest concentration of home births was in Lisbon, where the population is socially and economically more qualified (INE 2022), and it found correspondence with the already seen higher levels of education of the home birth mothers, as previously reported (Pintassilgo 2014).

The nationalities of the mothers who gave birth at hospitals were in accordance with the representativeness of these nationalities in the general set of women from foreign countries living in Portugal (Oliveira 2020). In the case of home births, European women emerged as one of the most-represented groups, which, to some extent, may be explained by the fact that in some European countries, home birth is better integrated with the overall delivery of maternity care (Galková et al. 2022).

The duration of pregnancy and parity also showed similarities in both contexts. The main dissonances were found between hospital and home births, as a whole, and births that took place elsewhere.

Considering the similarities of the mothers' profiles and pregnancy characteristics in both contexts (home and hospital births), a comparative analysis of the results between hospital and home births may have had the assistance dimension as a relevant and determining element. Nevertheless, the assistance analysis was hampered by the lack of information available in the official data on planned births and by the quality of the data, which was eventually compromised by the overvaluations of some categories and/or the invisibility of others. This dimension, which is influenced by planned or unplanned decisions, has impacted maternal and child health outcomes.

Thus, in considering birth assistance, the data found are surprising as the previous ethnographic work carried out in Portugal pointed to the significant presence of midwives providing home birth assistance (Santos 2020), which was contrary to these official statistics, which indicated that there was a higher presence of doctors. The existence of former qualitative research that mapped the characteristics of home births (a phenomenon with a relatively limited number of cases in Portugal) led to questioning the reliability of official data on birth assistance, as these records may overestimate the presence of doctors in this birth setting. International studies have reported the difficulty in having a good-quality database on home birth and its characteristics (Janssen et al. 2009; Galková et al. 2022), and in the case of Portugal, we argue that data on the type of professional birth attendant in home births should be used critically and with caution.

### 4.2. Home Birth and Health Outcomes

In Portugal, certain conditions of birth appear to be associated with the outcomes of this choice for women and families. Our analysis pointed to negative correlations between the frequency of home births assisted by trained healthcare professionals and the rates of perinatal, neonatal, and infant mortality, and we found a positive correlation between the frequency of home births with non-specialised care providers or unattended home births and infant mortality rates. With regard to these results, recently, the authors of the updated Cochrane review comparing planned hospital births with planned home births stated, "the evidence from randomised trials to support that planned hospital birth reduces maternal or perinatal mortality, morbidity, or any other critical outcome is uncertain" (Olsen and Clausen 2023).

Studies on other contexts have pointed to the level of integration of home births in the broader maternity care system as a key element for safety, with countries where home births are somewhat marginalised showing poorer perinatal outcomes associated with this place of birth (Snowden et al. 2015).

### 4.3. Home Birth and Maternity Care Systems

Having home births as part of the National Health Service appears to be too far away on the horizon in Portugal. With the (Circular 7495/2006), the then Minister of Health determined "the enshrinement of the right of each woman to freely choose the place where she wishes to give birth to her children with conditions of better quality for mother and child".

However, it is not clear if home birth could be considered under a strict interpretation of this Circular. It was published as part of the widely contested 2006–2007 national initiative for the centralisation of maternity care, which encompassed the closing of public maternity units with less than 1500 births per year to safeguard adequate levels of professional experience and expertise, but inevitably, it reinforced regional disparities (Matos 2010). The text of this Circular highlighted the progress made in the improvement of perinatal outcomes through hospital care, and as such, it conveyed the notion of the hospital being the only place where women may give birth safely. The right for women to choose the place of birth can be read here as their right to choose which hospital to give birth in, although this is not clearly stated (Santos 2020).

Integrating home birth care into the Portuguese health care system is likely to contribute to its safety, as has been pointed out by several authors considering other countries

(Hutton et al. 2019; Campbell et al. 2019; Comeau et al. 2018; Olsen and Clausen 2023; Quattrocchi, Patrizia 2014; Snowden et al. 2015) as it would potentially improve the quality of communication between home and hospital practitioners, promote timely emergency transport, increase the continuity of care, and reduce access inequalities caused by financial constraints. Instead of questioning the safety of home births, efforts should be made to question how to integrate home births in order to make them safer. In fact, previous studies have found no increased risk of adverse perinatal outcomes for planned home births among low-risk women in countries where home births are well-integrated into the maternity care system (De Jonge et al. 2015).

A recent guideline from the Society of Obstetricians and Gynaecologists of Canada (Campbell et al. 2019, p. 225) stressed that "[T]he data indicate that individuals at low risk for poor perinatal outcomes who plan homebirth with a regulated provider in an integrated health care system may have improved obstetric outcomes without increased neonatal morbidity or mortality. [...] In Canada, homebirth with a registered midwife or an appropriately trained physician is a reasonable choice for those who are evaluated to be at lower risk of obstetric or neonatal complications."

The integration of home birth care in the national health system would allow the systematic evaluation of quality (Santos 2020), ideally involving users and home birth professionals in all stages of policy-making and implementation. Across high-income countries, producing reliable data on the quality of maternity care—including women's assessments of the quality of care they received—and translating this evidence into public policies should be one of the main drivers of improving maternity care, moving to a women-centred paradigm and evidence-based care (Shaw et al. 2016). Creating further legal or regulatory limitations to home birth practice will not eliminate home births, as has been seen in European countries where home birth practice is illegal. Yet, forbidding or further marginalising home births definitely increases its risks, and this should not be the aim of any public policy.

### 4.4. Sources, Availability, and Quality of Data

Our analysis demonstrated how data on home birth should be read and analysed beyond numeric differences. Home births are not all the same, and statistical data, even if drawn from official records, are not immune to criticism. As in most European countries, Portuguese official statistics do not differentiate between planned and accidental home births or between planned hospital births and those that were planned to happen at home but were later transferred to a hospital. Yet, in 2010, a majority of all registered home births were from people from higher socioeconomic positions, suggesting that most registered home births were planned home births (Pintassilgo 2014). Similarly, our analysis showed that in 2020, most mothers of live births that occurred at home had higher and/or secondary education.

There were also some concerns regarding the accuracy of the official statistical data on the professional care providers in home births. Previous qualitative research contradicted the prevalence of doctors (to the detriment of nurses and midwives), which was revealed by the official statistics found for the last two decades. Nevertheless, distinguishing home births with specialised professional care providers from those with non-specialised care providers and those with no care providers rendered visible and important differences between them, underpinning the need to analyse home birth as a plural and non-homogenous phenomenon.

In Portugal, data about birth and perinatal outcomes are split through different sources of information, limiting the scope of data analysis. Hence, our analysis went as far as it could because the entries on foetal and neonatal deaths and the entries on live births were different. This lack of integrated information justified the correlation analysis we made in the second part of the article—between the proportion of births that occurred at home and the infant, perinatal, and neonatal mortality rates (with the last two considering foetal deaths).

As our analysis showed, a substantial percentage of births were recorded as having occurred in an "other place" (1.6% in 2020 compared to 1.3% of live births at home), with no other information regarding these occurrences. Despite gathering a considerable number of births, this category showed many missing data and unknown information about the characteristics of the births, the assistance, and the parents. In fact, the available social and clinical characteristics of the births that happened in an "other place" revealed important distinctions between these births and the ones that happened either at home or at a hospital (which were more similar to each other), which appears to be analytically relevant. The specificities of births in an "other place" deserve a dedicated in-depth analysis that can shed light on this place of birth.

Maternal mortality was left out of this analysis. However, the results recently published on maternal health in Portugal have shown a worsening of the maternal mortality rate, which has reached values of approximately 20 maternal deaths per 100,000 births in 2020 (Pintassilgo et al. 2022). The contribution of home births to this increase is uncertain, demanding further analysis which should be qualitative, quantitative, multidimensional, and multidisciplinary. There have been examples of good practices in monitoring maternal outcomes considering different conditions of birth, in different countries, namely, in the European context (Birthplace in England Collaborative Group 2011; De Jonge et al. 2013; INSERM 2021).

## 5. Conclusions

This analysis of home births in Portugal showed that it is not a homogeneous phenomenon. By distinguishing home births by the type of assistance, those with specialised care providers were negatively correlated with perinatal, neonatal, and infant mortality, and home births with non-specialised professional care providers or those that were unassisted showed positive correlations with these mortality rates. According to our analysis, in Portugal, planned home birth with the assistance of a midwife, obstetric nurse, or doctor does not appear to determine worse perinatal health outcomes. In fact, in 2020, the social profiles of the mothers, the lengths of the pregnancies, the levels of parity, and the types of assistance for home births were generally more similar to those of hospital births than to the characteristics of births occurring in an "other place". Although demanding further investigation, the analysis pointed to need to better integrate home births in the health system and in official statistical records in order to assure a systematic evaluation of the quality of care that distinguishes planned home births from accidental home births and to consistently improve the safety of home births for families and professionals.

**Author Contributions:** Conceptualization, D.M.N., M.J.D.S.S. and S.P.; methodology, I.T. and S.P.; software, I.T. and S.P.; validation, D.M.N., I.T., M.J.D.S.S. and S.P.; formal analysis, I.T. and S.P.; investigation, D.M.N., M.J.D.S.S. and S.P.; resources, D.M.N., I.T., M.J.D.S.S. and S.P.; data curation, S.P.; writing—original draft preparation, M.J.D.S.S. and S.P.; writing—review and editing, D.M.N., I.T., M.J.D.S.S. and S.P.; visualization, M.J.D.S.S. and S.P. All authors have read and agreed to the published version of the manuscript.

**Funding:** This work was supported by the Fundação para a Ciência e a Tecnologia (FCT) through the funding of the R&D Unit UIDB/03126/2020 and SFRH/BD/99993/2014 (M.J.D.S.S.).

**Institutional Review Board Statement:** The study was conducted according to the guidelines of the Declaration of Helsinki and the code of conduct of Ethics of Iscte—Instituto Universitário de Lisboa, 2016, in the general framework of the mission and duties of the Ethics Committee of Iscte (Order No. 7095/2011; Diário da República, 2.ª série—N.º 90—10/06/2011).

**Informed Consent Statement:** Secondary and anonymized data used in the study were produced by the National Institute of Statistics, according to the principles of the National Statistical System (Law No. 22/2008, 13 May).

**Data Availability Statement:** Data was accessed under request within the scope of the existing protocol between INE and the Ministry of Science and Higher Education, which allows accredited researchers to access official statistical information made available through microdata.

**Conflicts of Interest:** The authors declare no conflict of interest.

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
