# Peer review of "Home Birth in Portugal—A Comprehensive Analysis Based on Official Statistical Data"

_socsci, doi:10.3390/socsci12060314_

Round 1

Reviewer 1 Report

This is a paper on planned home birth in Portugal and could add a new jurisdiction to the growing body of global literature on outcomes and characteristics of planned home birth – no previous analyses of place of birth based on large population data derived from the Portugal national vital statistics database has been published. Specifically, this paper proposes to 1) describe the characteristics of people give birth in Portugal by planned place of birth and (2) to analyze the association between home birth 7 assistance and perinatal, neonatal, and infant mortality rates.  To some extent these objectives are achieved; however, there are both minor and moderate issues that will need to be addressed to help this paper to make an important and credible contribution:

Moderate revisions required:

The paper is clearly written with respect to English and syntax; but some editing throughout is necessary to neatly tie the objectives to measures, methods, results, and discussion. As it is, there appears to be several extraneous data points and analyses that are included without a clear rationale and sometimes no mention in the discussion. 

·       For example, the title suggests that the analysis will look at before and after pandemic incidence and/or outcomes, yet the reference to the impact of Covid on incidence is only peripherally mentioned in the paper in Lines 95 and 168 and nothing is presented about analysis of outcomes related to the pandemic.  Suggest changing the article title at minimum.

·       Similarly, Lines 88-92 say that maternal mortality was left out as a factor for analysis– correctly noting that the incidence is so low as to prevent a nuanced quantitative assessment, yet then in Figure 4 and Lines 285 the authors include Maternal Mortality Rate in a supposed correlation analysis/graph!!  Recommended removing that variable from the graph if it cannot be clearly linked to place of birth.

·       Lines 146-158 on fertility rates in Portugal, while interesting, do not seem relevant to the topic at hand

·       The relevance of the individual factors that the authors use to describe “characteristics of people” are not well described or supported in the background section – appears to be simply the reporting of factors available through the national administrative database. Also discussion is thin on why differences found in nativity, education, or insurance might matter or relevance to outcomes reported such as perinatal mortality or

Literature review/background

·       64-70  Authors state  It is noteworthy that even in countries where data on birth outcomes by place of birth 64 are available, the existing literature does not always detail the conditions in which these 65 births take place, as if they were homogeneous or essentially defined by the place itself - 66 home or hospital. “  Not strictly true: Since 2000,  there is a robust high quality body of published research on the sociodemographic characteristics of people who choose and/or can access community birth, on impact of professional attendants,  as well as on the impact of interprofessional collaboration on integration of midwives and planned home births  on outcomes.   I refer the authors to Janssen 2002, 2009 (organization and outcomes) and 2009 (patient characteristics), Hutton 2009-2014 (outcomes in the context of midwife-led integrated care), de Jonge (very large, population data) Birthplace In England (2011),  Cheyney (2014), Scarf (2018, 2019), Olson (Cochrane) 2012, Vedam (on quality of birth certificate data and provider identification 2003, and transfer and access to skilled attendants and integrated care, 2014) and MacDorman 2011, 2014, 2022, trends, sociodemographic characteristics, access to funded home birth by race, and types of providers/birth attendants.  Interestingly in Lines 359-361 authors cite the articles that should be examined and discussed in literature review, and in the bibliography some others appear (de Jonge, Birthplace in England) but they are not cited in the body of the article.

It would be better to discuss the findings of the above literature on place of birth in high resource countries and then in the discussion refer back to how the findings in Portugal align or diverge from the published literature.  It is enough to say this analysis is the first of its kind in the country.

·       Background: Needs more information about the organization of maternity care and what types of providers attend home births in Portugal currently.   Lines 336-343 could be moved to background and some information about types of provider available to attend in each setting, and covered by national or private insurance, would be helpful and informative

·       Needs some serious citation review and management  - For example 3 important and relevant articles by de Jonge and Birthplace In England collaborative group appear in the reference list but are not linked to any sentence in the article.

Methods

·       Reporting of data sources very clear and transparent. 

·       Definitions provided are clearly stated, but rationale for inclusion of each data point not always clear. Suggest clearly reporting how variables/metrics for analysis that were selected link to the objectives of the paper.

·       Correlation analysis and data graphs are mostly clearly presented, but might benefit from showing the trends with an overlay line for those readers who are not used to reading such graphs.  Also discussion of findings should link clearly to each datapoint examined, with relevance to the research question explicated. Perhaps subheadings would help.

Discussion

Without the clarity of how measures link to hypotheses or research questions and summary findings (eg that the analysis demonstrated no significant differences in key outcomes by place of birth when attended by professional attendants), the discussion currently reads more like an implications section with some editorializing. Suggest using some of the same subheadings in the methods section.

Minor

·       Table 2 and throughout – suggest using the term Care Provider instead of assistance – more aligned with other published literature on planned home birth characteristics and outcomes.

·       “planned home birth” please report how you are defining this in your analysis and align with published literature

·       “Home birth assistance” – define – skilled birth attendant? Professional attendant? Doctor or Midwife attendant? How verified?

·       Lines 197-202 why is fathers education important? Please provide a rationale for inclusion, discuss implications, or remove to save space

·       Line 265, indeed surprising finding and authors are correct to question veracity of data  – official records of birth frequently incorrectly record the person who verifies the birth on vital stats, birth certificate data see: Review of Pang study:  Vedam, Birth:Issues in Perinatal Care 2003

Author Response

We would like to thank the comments of the reviewers and the opportunity to further improve our work. We have carefully revised our manuscript according to the reviewers’ comments and recommendations. Next, we provide detailed responses to the reviewers’ comments (in bold).

 Reviewer 1

This is a paper on planned home birth in Portugal and could add a new jurisdiction to the growing body of global literature on outcomes and characteristics of planned home birth – no previous analyses of place of birth based on large population data derived from the Portugal national vital statistics database has been published. Specifically, this paper proposes to 1) describe the characteristics of people give birth in Portugal by planned place of birth and (2) to analyze the association between home birth 7 assistance and perinatal, neonatal, and infant mortality rates.  To some extent these objectives are achieved; however, there are both minor and moderate issues that will need to be addressed to help this paper to make an important and credible contribution:

We thank for the encouraging comments regarding some aspects of our work.

Moderate revisions required:

The paper is clearly written with respect to English and syntax; but some editing throughout is necessary to neatly tie the objectives to measures, methods, results, and discussion. As it is, there appears to be several extraneous data points and analyses that are included without a clear rationale and sometimes no mention in the discussion. 

  • For example, the title suggests that the analysis will look at before and after pandemic incidence and/or outcomes, yet the reference to the impact of Covid on incidence is only peripherally mentioned in the paper in Lines 95 and 168 and nothing is presented about analysis of outcomes related to the pandemic. Suggest changing the article title at minimum.

We thank the reviewer for signal the absence of analysis of outcomes related to the pandemic, as suggested in the tittle. We agree with the observations and the article title was changed, accordingly (please check for changes made in the manuscript: Title). Also, the text mentioning Covid impact in results on birth assistance has been reformulated (please check for changes made in the manuscript: Introduction, lines 106-108 and 186-187).

  • Similarly, Lines 88-92 say that maternal mortality was left out as a factor for analysis– correctly noting that the incidence is so low as to prevent a nuanced quantitative assessment, yet then in Figure 4 and Lines 285 the authors include Maternal Mortality Rate in a supposed correlation analysis/graph!! Recommended removing that variable from the graph if it cannot be clearly linked to place of birth.

      We agree with the reviewer that the inclusion of maternal mortality rate in graph 4 is inconsistent with not considering it for further analysis. Accordingly, we have removed the indicator from Figure 4, as suggested (please check for changes made in the manuscript: Results, line 303 and Figure 3).

  • Lines 146-158 on fertility rates in Portugal, while interesting, do not seem relevant to the topic at hand

      Considering this recommendation, we removed the section 3.1. from Results section (please check for changes made in the manuscript: Results, lines 164-176).

  • The relevance of the individual factors that the authors use to describe “characteristics of people” are not well described or supported in the background section – appears to be simply the reporting of factors available through the national administrative database. Also discussion is thin on why differences found in nativity, education, or insurance might matter or relevance to outcomes reported such as perinatal mortality or

      We acknowledge the question raised. We have introduced evidence from previous studies concerning the importance of parents characteristics in care assistance (please check for changes made in the manuscript: Introduction, lines 84-86) and introduced related considerations in discussion (please check for changes made in the manuscript: Discussion, lines 364-…).

Literature review/background

  • 64-70 Authors state  “It is noteworthy that even in countries where data on birth outcomes by place of birth 64 are available, the existing literature does not always detail the conditions in which these 65 births take place, as if they were homogeneous or essentially defined by the place itself - 66 home or hospital. “ Not strictly true: Since 2000,  there is a robust high quality body of published research on the sociodemographic characteristics of people who choose and/or can access community birth, on impact of professional attendants,  as well as on the impact of interprofessional collaboration on integration of midwives and planned home births  on outcomes.   I refer the authors to Janssen 2002, 2009 (organization and outcomes) and 2009 (patient characteristics), Hutton 2009-2014 (outcomes in the context of midwife-led integrated care), de Jonge (very large, population data) Birthplace In England (2011),  Cheyney (2014), Scarf (2018, 2019), Olson (Cochrane) 2012, Vedam (on quality of birth certificate data and provider identification 2003, and transfer and access to skilled attendants and integrated care, 2014) and MacDorman 2011, 2014, 2022, trends, sociodemographic characteristics, access to funded home birth by race, and types of providers/birth attendants.  Interestingly in Lines 359-361 authors cite the articles that should be examined and discussed in literature review, and in the bibliography some others appear (de Jonge, Birthplace in England) but they are not cited in the body of the article.

It would be better to discuss the findings of the above literature on place of birth in high resource countries and then in the discussion refer back to how the findings in Portugal align or diverge from the published literature.  It is enough to say this analysis is the first of its kind in the country.

We thank very much for these comments and references. We recognize the body of knowledge produced on these matter may take into account sociodemographic characteristics or type of professional attendance. Yet these are not directly used as means of demonstrating the insufficiency of the variable “place of birth” for the analysis of planned home birth vs. planned hospital birth. They are mainly used to describe the study sample/population.

Yet, considering the suggestions, we have integrated references in the text and renewed the research done, based on the references indicated.

  • Background: Needs more information about the organization of maternity care and what types of providers attend home births in Portugal currently. Lines 336-343 could be moved to background and some information about types of provider available to attend in each setting, and covered by national or private insurance, would be helpful and informative

We thank the comment and suggestion, and we agree with the reviewer. We added the information in section of Introduction (please check for changes made in the manuscript: Introduction, lines 49-57).

  • Needs some serious citation review and management - For example 3 important and relevant articles by de Jonge and Birthplace In England collaborative group appear in the reference list but are not linked to any sentence in the article.

      We reviewed the bibliography used and presented in the references.

Methods

  • Reporting of data sources very clear and transparent.

      We thank for this comment.

  • Definitions provided are clearly stated, but rationale for inclusion of each data point not always clear. Suggest clearly reporting how variables/metrics for analysis that were selected link to the objectives of the paper.

      We agree with the reviewer and have included the link between variables/metrics and the objectives of the paper (please check for changes made in the manuscript: Materials and Methods, lines 139-141, 160-162).

  • Correlation analysis and data graphs are mostly clearly presented, but might benefit from showing the trends with an overlay line for those readers who are not used to reading such graphs. Also discussion of findings should link clearly to each datapoint examined, with relevance to the research question explicated. Perhaps subheadings would help.

      We thank for the comment and follow the suggestion, by introducing trend lines into the charts (please check for changes made in the manuscript: Results, Figures 4 to 6). We also deepened the discussion regarding these results (please check for changes made in the manuscript: Discussion, lines 406-421).

Discussion

Without the clarity of how measures link to hypotheses or research questions and summary findings (eg that the analysis demonstrated no significant differences in key outcomes by place of birth when attended by professional attendants), the discussion currently reads more like an implications section with some editorializing. Suggest using some of the same subheadings in the methods section.

We agree with the reviewer and have included some subheadings in the discussion, accordingly previous sections (please check for changes made in the manuscript: Discussion).

Minor

  • Table 2 and throughout – suggest using the term Care Provider instead of assistance – more aligned with other published literature on planned home birth characteristics and outcomes.

      We thank the suggestion and made the replacement, as suggested (please check for changes made in the manuscript: Material and Methods, and Results, Table 2, (…)).

  • “planned home birth” please report how you are defining this in your analysis and align with published literature

We thank the suggestion. A definition was added (please check for changes made in the manuscript: Introduction, lines 57-58).

  • “Home birth assistance” – define – skilled birth attendant? Professional attendant? Doctor or Midwife attendant? How verified?

We thank the comment. When we referred to "home birth assistance", we were not defining a priori this assistance, which in fact could be either differentiated/skilled/professional or not. This definition was not up to us, but was based on official records on the presence or absence of specialized assistance during childbirth and the professional category of the care provider who attended the birth. We expect replacing the term “assistance” with “care provider” made this clearer.

  • Lines 197-202 why is fathers education important? Please provide a rationale for inclusion, discuss implications, or remove to save space

Considering this recommendation, we removed the information concerning fathers’ education (please check for changes made in the manuscript: Material and Methods, line 136; Results, Table 2, lines 215-221).

  • Line 265, indeed surprising finding and authors are correct to question veracity of data – official records of birth frequently incorrectly record the person who verifies the birth on vital stats, birth certificate data see: Review of Pang study:  Vedam, Birth:Issues in Perinatal Care 2003

The references have been updated, considering the suggestion.

Reviewer 2 Report

I have thoroughly enjoyed reading this article. It is well put together and extremely well written. A few minor typos/edits required eg number such as 1950s do not need apostrophes. The discussion paragraph has sentences beginning with conjunctions (plus popped up a lot). These need removed.  Otherwise it is excellent. 

Author Response

Reviewer 2

I have thoroughly enjoyed reading this article. It is well put together and extremely well written. A few minor typos/edits required eg number such as 1950s do not need apostrophes. The discussion paragraph has sentences beginning with conjunctions (plus popped up a lot). These need removed.  Otherwise it is excellent. 

 Thank you very much for the review of the article and for the appreciations. We agree with the suggestions and have made changes to the text accordingly (please check for changes made in the manuscript, namely in Discussion).

Round 2

Reviewer 1 Report

I have reviewed the revisions and they are acceptable to me - to accept in this revised form.

Author Response

We thank the reviewer for the careful reading of the manuscript and for the comments at different stages of the process, which helped us to improve the final result.